# Photon-Counting Computed Tomography (PCCT): Technical Background and Cardio-Vascular Applications

**DOI:** 10.3390/diagnostics13040645

**Published:** 2023-02-09

**Authors:** Antonella Meloni, Francesca Frijia, Daniele Panetta, Giulia Degiorgi, Carmelo De Gori, Erica Maffei, Alberto Clemente, Vincenzo Positano, Filippo Cademartiri

**Affiliations:** 1Department of Radiology, Fondazione G. Monasterio CNR-Regione Toscana, 56124 Pisa, Italy; 2Unità Operativa Complessa di Bioingegneria, Fondazione G. Monasterio CNR-Regione Toscana, 56124 Pisa, Italy; 3Institute of Clinical Physiology, National Council of Research, 56124 Pisa, Italy

**Keywords:** photon-counting detectors, photon-counting CT, cardiac CT, CT angiography

## Abstract

Photon-counting computed tomography (PCCT) is a new advanced imaging technique that is going to transform the standard clinical use of computed tomography (CT) imaging. Photon-counting detectors resolve the number of photons and the incident X-ray energy spectrum into multiple energy bins. Compared with conventional CT technology, PCCT offers the advantages of improved spatial and contrast resolution, reduction of image noise and artifacts, reduced radiation exposure, and multi-energy/multi-parametric imaging based on the atomic properties of tissues, with the consequent possibility to use different contrast agents and improve quantitative imaging. This narrative review first briefly describes the technical principles and the benefits of photon-counting CT and then provides a synthetic outline of the current literature on its use for vascular imaging.

## 1. Introduction

Vascular diseases refer to any abnormal condition that affects the blood vessels. The most important role among vascular diseases is played by atherosclerosis. The atherosclerotic plaques, composed by fatty substances, cholesterol, cellular waste products, calcium, and fibrin [1,2], can cause the narrowing or hardening of arteries, with consequent ischemia and functional impairment of the affected tissue or organ. Indeed, atherosclerosis is the leading cause of death and disability around the world [3]. The diagnosis of the ischemic disease at an early stage is the key to improve the effectiveness of patient treatment and to implement additional preventive measures, impeding its tragic sequelae.

Computed tomography (CT) has gained a pivotal role in the assessment of vascular disease processes and in the planning and follow-up of minimally invasive interventions. In particular, coronary CT angiography (CCTA) has progressively gained widespread adoption during the last two decades [4], thanks to increased axial coverage of multi-row detectors, increased rotation speed, and progress in prospective gating protocols based on electrocardiogram (ECG) phase-correlated triggering [5]. The main advantages of CT are its non-invasiveness, large availability, fast scanning speed, wide field of view, and excellent spatial and temporal resolution [6]. The main limitations of CT are the exposure to ionizing radiations and the necessity of administering iodinated contrast media, which can be problematic for patients with kidney disease [7].

Photon-counting computed tomography (PCCT) is a technology based on energy-resolving, direct-conversion X-ray detectors, which has been adopted just very recently in clinical CT equipment after 15 years of research and development. This is substantially different from conventional CT detectors, based on indirect X-ray conversion (with scintillators) and signal integration over the entire X-ray energy spectrum. PCCT conveys the potential for changing the clinical CT scenario, thanks to its many inherent advantages and the ability to overcome several of the shortcomings of current state-of-the-art CT systems.

The goals of this narrative review are to describe the technical principles of PCCT, to outline its benefits over conventional CT technology, and to present the current vascular applications.

## 2. Search Strategy

To prepare the narrative review we followed the indications present in [8]. The article search was performed on PubMed, Scopus, and Google Scholar electronic databases between August and October 2022. We used the keywords “photon-counting computed tomography”, “PCCT”, “photon counting detector”, “photon counting X-ray detectors”, “photon counting CT”, and “spectral CT”. Only articles written in English were included. Additional records identified through the list of references or other sources were also included. Two reviewers (AM and FF) analyzed the scientific papers to extract the relevant data for the purpose of this work.

## 3. Photon-Counting Detector Technology

### 3.1. Comparison between Conventional and Photon Counting Detectors

The X-ray detector is a core component of a CT scanner, determining both image quality and radiation dose. Conventional CT devices currently employ energy-integrating detectors (EIDs) equipped with scintillator elements and reflective layers (septa). The layer of ceramic scintillators converts the incident X-ray photons into low energy secondary photons in the visible spectrum. These latter photons are then absorbed by a photodiode array made of a semiconducting material, which generates an electrical signal proportional to the total deposited energy, summed to electronic thermal noise. Finally, the electrical signal is amplified and then converted to a digital signal, so that it can be processed for tomographic image reconstruction. Indeed, because the detector integrates the energy from all incident photons in a given time interval, any information regarding the energy of an individual X-ray photon is lost. Septa are incorporated between scintillating detector pixels to prevent light crosstalk between them. Such septa cause “dead space” on the detector surface and, since there is a physical limit to their thickness, they limit the geometric dose efficiency [9,10].

PC detectors (PCDs) are based on a direct conversion technique. They are made by coupling a semiconductor sensor with a high effective atomic number (typically, cadmium telluride or CdTe) and high thickness (1–2 mm) with a readout circuit (Figure 1). Upon absorption in the semiconductor, the incident X-ray photons are directly converted into electron-hole pairs. Charge collection efficiency is increased by applying an inverse bias voltage through either Schottky-type barrier contact or ohmic (metal semiconductor) contacts, thus improving the energy resolution, detection efficiency, and reducing the contribution of dark current [11]. When operated in counting mode, the height of the electric pulse is proportionate to the energy deposited by the interacting X-ray photon in the depleted semiconductor region. Pulse heights are then compared with a voltage that reflects a specified photon energy level (energy threshold) [12]. Multiple electronic comparators are used to count the number of pulses with an energy level equal to or greater than the preset thresholds [13], allowing us to sort the incoming photons into a number of energy bins (typically two to eight). In experimental settings, the energy thresholds (in kiloelectron volts, or keV) could be set up by the user before data acquisition; on the other hand, commercial systems are generally pre-set to factory values and do not allow the final user to create custom thresholds. The lower threshold is set at levels that are reasonably higher than the electronic noise level, so that the electronic noise is totally suppressed in the final signal. The other thresholds are either spaced uniformly or chosen to optimize a given imaging task [14].

### 3.2. Technical Challenges of PCDs

The adoption of PCDs in clinical CT has been made possible by the minimization or overcoming of several factors affecting the detector performance and, in turn, the final quality of reconstructed images. Count rate performance was one of the main obstacles to the introduction of counting mode CdTe detectors in clinical CT. Photon fluences in commercial scanners can be higher than 108 photons/(mm^2^ s) [15], which is several orders of magnitudes as compared to hit rates normally encountered in, for instance, nuclear medicine applications. In clinical PCDs, the requirement for high hit-rate-capable detectors (>106 counts per second) can be relaxed by using monolithic CdTe layers, bonded to pixelated application-specific integrated circuits (ASIC) with small pitch (<200 m). As an example, in 2021 Siemens Healthineers released the first commercial CT scanner based on PCD, with FDA clearance for clinical use. In this case, the pitch of the ASIC reading the CdTe detector was 0.150 × 0.176 mm^2^ at the isocenter, with standard reading after 2 × 2 binning. Reducing the active area for each pixel, and consequently also the count rate requirement, pulse pileup, and consequent count loss and spectral distortion, this configuration leads to a very high spatial resolution when operating the detector without rebinning (i.e., 1 × 1 reading) [16]. On the other hand, narrower detector pitch leads to secondary charge clouds being sensed by more than one neighboring pixels, which is known as charge sharing effect [17]. Charge sharing can be corrected by several schemes, such as dedicated circuits implementing a winner-takes-all strategy, where the pixel receiving the largest amount of charge is assigned to the total charge detected in a 2 × 2 neighborhood [18]. The Medipix3 ASIC, for instance, implements this strategy [19]. Besides secondary charge sharing, other types of pixel crosstalk are possible in PCDs, such as those related to the generation of fluorescence X-ray radiation in pixels involved in the first interaction, which is in turn detected in the nearby pixels. This effect is responsible for the lower limit of detector pixel size in practical applications [13].

## 4. Benefits of PCDs

This section describes the advantages of the PCCT system over conventional energy-integrating CT.

### 4.1. Reduction of Electronic Noise

In CT the electronic noise is mainly caused by the analog electronic circuits in the detection system and is usually detected as a low-amplitude signal.

EIDs do not process the signals from individual photons but integrate the total energy deposited over a certain time period, including electronic noise. Conversely, in PCDs the noise affects the minimum detectable pulse-height (noise threshold) but not the number of pulses (photon count) above that threshold. Therefore, by setting up the low energy threshold of a PCD at levels exceeding the one associated with the noise floor (approximately 25 keV), the electronic noise can be effectively excluded from photon and/or pulse counts, although it is still present in the spectral information [13].

Therefore, for the same dose, the noise in the reconstructed image is lower with PCDs than with EIDs. The intrinsic advantage of PCDs is of particular benefit in CT scans performed at very low radiation dose or in obese patients, when the noise is not negligible. In these scenarios, images obtained with PCDs have demonstrated to be less affected by streak artifacts and signal uniformity and to produce more stable CT-numbers [20,21]. Importantly, the reduced noise benefit can be exploited to improve the dose efficiency, since with PCDs a noise level comparable to that of an EID can be obtained with a lower radiation dose.

### 4.2. Improvement in Spatial Resolution

In current clinical EIDs the pixel size is about 0.4–0.6 mm at the isocenter, limiting their resolution. In fact, the design of smaller detector pixels causes an increase in the relative area to the septum in comparison to the detector area, with a consequent reduction in the geometric dose efficiency. In PCDs, thanks to the absence of a mechanic separation, the pixel pitch does not have a technical limitation and can reach 0.15–0.225 mm at the isocenter [22,23,24,25].

Different solutions have been proposed to improve the spatial resolution of scintillator detectors. For example, a dedicated attenuating filter can be used to decrease the pixel aperture [26]. However, this approach reduces the radiation dose efficiency. An ultra-high-resolution (UHR) EID-CT, characterized by a thinner septa gap, has been introduced to the market [27]. This system has an effective detector pixel size of 0.25 × 0.25 mm, which reduces the gap with the PCDs. Anyway, a UHR imaging technique has been developed also for PCD-based CT systems [28]. A study conducted on anthropomorphic phantoms and cadaveric specimens has shown that UHR PCD images have 29% lower noise compared to the UHR EID images, which can be translated into a potential dose savings of 50% for equivalent image noise [28].

### 4.3. Contrast Improvement

Conventional EIDs weigh the photons based on the energy, so that the contribution to the signal is higher for high-energy than for low-energy photons. Since low-energy photons carry more contrast information than high-energy photons, their underweighting reduces the contrast-to-noise ratio (CNR). Moreover, the non-uniform weighting of photons increases the variance relative to the mean value, resulting in a reduced signal-to-noise ratio (Swank factor) [29].

Conversely, in PCDs all photons are equally weighted, independent from the energy (one photon, one count). The possibility to give relatively more weight to low-energy photons translates into a higher contrast in comparison with EIDs, in particular for low absorbing materials [30,31,32,33]. Importantly, the CNR for a given material can be maximized by tailoring the weighting scheme, that is by giving different weights to photons of different energy [16,34,35]. PCDs also avoid the Swank factor.

### 4.4. Reduction of Beam-Hardening 

CT uses polyenergetic beams and, due to the energy dependence of mass attenuation coefficients, low-energy photons are more attenuated than high-energy photons [36]. This causes a shift of the mean energy of the X-ray beam toward the higher end of the spectrum, a phenomenon known as beam hardening. Beam hardening from a very dense target (i.e., cortical bone and metal implants) may result in characteristic artifacts: cupping artifacts and streaking (dark bands) artifacts [37]. These artifacts affect the image appearance and CT number accuracy for nearby soft tissues.

In PCDs the constant weighting reduces the beam-hardening artifacts [38,39]. In particular, in PCDs the best advantages in terms of immunity to beam-hardening effects are obtained with the use of high-energy thresholds [20,40].

### 4.5. Multienergy Acquisitions and K-Edge Imaging

Spectral CT refers to the use of energy-dependent attenuating characteristics of materials for achieving differentiation between tissue compositions (material decomposition) and improved lesion detection in contrast-enhanced scans and for reducing imaging artifacts [41].

Currently, material decomposition is performed using EID-based CT scanners capable of acquiring (with different approaches) dual-energy data, that are then combined through automated or semi-automated post-processing [42,43]. However, the conventional scanners do not allow for the separation of more than two materials (or three, by using some prior information and/or additional conditions) [44], require temporal image registration, and suffer from spectral overlap, which reduces the accuracy of material decomposition [45,46].

Since PCD can discriminate photons of different energies through pulse-height analysis, they inherently allow simultaneous multi-energy (*n* ≥ 2) acquisitions, with perfect spatial and temporal registrations and without spectral overlap [47]. Importantly, PCDs with more than three bin counters can allow to characterize the composition of each voxel as a combination of three or more basis materials [48].

The implementation of material decomposition algorithms from a number of energy-selective images results in a set of basis image maps that show the distribution within the imaged object of a certain material. These basic images can be processed to obtain virtual monochromatic images (VMI) [49,50,51], virtual non-contrast images [52], or material-specific color-overlay images [53]. PCCT provides several benefits over conventional EID-based CT scanners for these purposes. In addition to the comparable accuracy in iodine quantification and VMI CT number, PCCT offers the advantages of temporal and spatial alignment to avoid motion artifact, high spatial resolution, and improved contrast-to-noise ratio [50]. Due to its improved spectral separation, PCCT can foster more realistic virtual non-contrast images [50]. Moreover, with PCDs the higher number of energy measurements results in a more accurate measurement of each photon energy and, of consequence, in lower image noise in the material-specific images [54]. Importantly, PCDs permit to improve the accuracy for measurement of the concentration of a contrast agent. In fact, the techniques used to measure water, calcium, and contrast agent (i.e., iodine) with dual-energy-CT (DECT) require assumptions about the tissue composition [45] and if these assumptions are not correct, the measurement becomes inaccurate [55].

A key application of spectral/multi-energy CT is K-edge imaging, that is, the imaging of materials with a detectable K-edge in the diagnostic X-ray energy range. K-edge imaging is not feasible with DECT, while PCCT enables it, in virtue of the possibility of selecting energy thresholds lower and higher than the K-edge of a specific material. The detectability of a K-edge depends on the finite widths of the energy bins and then images can be reconstructed from the corresponding projections in the two energy bins. Different contrast agents can be distinguished according to their unique K-edges, besides the similar Hounsfield numbers in conventional CT images [56,57,58,59]. Therefore, PCTT provides the unique chance to use different contrast agents from iodine, including gold, platinum, silver, ytterbium, and bismuth, that are within the clinical X-ray tube spectrum [58,60,61,62], and to develop new types of contrast agents including nanoparticles targeted to specific cells or enzymes [59,63,64,65]. This approach opens the doors to molecular and functional CT imaging and to the simultaneous administration and detection of specific distribution of different contrast agents [53,66,67,68], conveying additional information. However, the promising results from animal or proof-of-concept (in silico) studies have not yet been translated into clinical practice. 

### 4.6. Dose Efficiency

Thanks to the reduction of the electronic noise and the improvements in CNR and visualization of small objects, PCDs lead to better dose efficiency than EIDs [21,69]. This makes PCCT a promising technique for new low-dose imaging protocols, protecting patients from high radiation dose exposure while maintaining good image quality

## 5. Pre-Clinical and Clinical Studies

Several phantom, animal, and even human studies have been conducted in recent years to evaluate the potential of PCCT as a more-performant alternative to conventional CT. This section provides an overview of the existing literature focused on vascular applications (Table 1). 

### 5.1. Coronary Imaging

CCTA is the favored imaging modality for the non-invasive assessment of coronary artery disease (CAD), allowing us to visualize the coronary lumen, assess stenosis, and identify plaque features in three dimensions [70] (Figure 2). In this context, the high or ultra-high spatial resolution, achievable with both UHR EID-CT or PCCT scanners, could be particularly beneficial. Indeed, it conveys the potential for a more comprehensive evaluation of the coronary tree, a more precise grading of stenosis, and a better evaluation of segments with stents or extensive calcifications [71].

The largest published clinical study of PCCT for CAD, involving 92 patients, demonstrated excellent imaging quality, a very high CNR, and a good ability to assess coronary segments and vessels, even in cases of calcified plaques and stents [72]. Indeed, only 5% of the segments were rated non-diagnostic. The radiation dose was generally low and depended strongly on the scan mode. Nine patients also underwent invasive coronary angiography as reference standard and the PCCT showed very high diagnostic performance for significant CAD on a per segment level (sensitivity 92% and specificity 96%).

Accurate quantification of stenosis severity is paramount for planning a correct therapeutic approach [73] and CT angiography (CTA) enables the measurement of percent area stenosis [74]. Current CTA-based stenosis measurements rely primarily on the segmentation of the iodinated lumen and work for both circular and noncircular transverse luminal profiles. However, calcification produces blooming and partial volume artifacts on CT imaging, which can prevent the accurate evaluation of the coronary artery lumen and result in an overestimation of the stenosis, determining a false positive diagnosis [75,76]. In addition, calcifications and the iodinated lumen may have similar attenuation properties. Li et al. proposed a method to determine the percent area of stenosis that uses material decomposition of dual-energy and multiple-energy CT images and does not require segmentation [77]. Computer simulation demonstrated that this method was able to reduce partial volume and blooming effects while phantom experiments showed accurate and reproducible stenosis measurements from multiple-energy CT images. Importantly, for four-threshold PCCT images, the estimation errors were lower than for DECT and two-threshold PCCT images, and the three-basis-material decomposition performed directly on them generated calcium, iodine, and water maps.

CT-based coronary plaque characterization and quantification may help to identify patients at risk for future adverse cardiac events [78,79]. However, conventional CCTA still suffers from a limited spatial resolution and soft-tissue contrast, which impairs its diagnostic performance for high-contrast (calcification) and low-contras (noncalcified plaque) tasks. Boussel et al. scanned with PCCT 10 calcified and 13 lipid-rich non-calcified plaques from post-mortem human coronary arteries, demonstrating the ability of this technique to differentiate between the normal wall, the lipid-rich plaque, the calcification, and the surrounding adventitial and perivascular fat, based on differences in spectral attenuation and iodine-based contrast agent concentration [80]. Si-Mohamed et al. compared the quality of CCTA scans obtained with PC and conventional dual-layer CT systems in 14 patients with CAD [81]. According to the five-point score analysis performed by three experienced cardio-radiologists, significant improvements in overall image quality, diagnostic quality, and diagnostic confidence were obtained with PCCT, in both calcified and non-calcified plaques. Their findings were further confirmed by the phantom study where PCCT images showed, in comparison to EID-based CT images, a 2.3- and 2.9-fold increased detectability index for coronary lumen and non-calcified plaque, respectively. Mergen et al. scanned with PCCT 20 patients with atherosclerotic plaques in proximal coronary arteries and tested three different slice thicknesses and two kernels reconstructions, evaluating atherosclerotic plaques with a semiautomatic software after a manual definition of coronary arteries by an expert radiologist [82]. Their results showed significantly lower total plaque volumes and significantly lower calcified plaque components but a higher volume of noncalcified plaque components on ultra-high-resolution CT, suggesting that higher spatial resolution results in reduced blooming artifacts and better visualization of non-calcified plaque components.

The amount of coronary artery calcium (CAC), quantified according to the Agatston methodology, has been demonstrated to be an independent predictor of cardiovascular events and has been proposed as a screening tool for CAD in asymptomatic subjects [83]. However, with conventional CT systems, the accuracy of CAC quantification is affected by blooming artefacts around CAC and partial volume effects preventing to detect small CAC. Moreover, the benefits of screening need to be weighed against the risks associated with ionizing radiations in asymptomatic individuals. Studies conducted in both phantoms [84,85] and cadaveric specimens [85,86] demonstrated that the inherent advantages of PCDs over the conventional EIDs could be effectively translated into a more accurate and reproducible coronary calcium detection and quantification. Importantly, in phantoms PCCT maintained and improved CAC detection even at 50% radiation dose and accurately measured physical volumes, especially at reduced slice thickness and for high-density CAC [84]. The same group determined mono-energetic (monoE) level-specific Agatston score thresholds for CAC scoring on PCCT and demonstrated in phantoms that, thanks to an increased CNR, virtual monoE images at low energy levels allowed for a radiation dose reduction of 50% for medium- and high-density CAC [87]. The positive impact of PCCT on CAC scoring has also been tested in human studies. Symons et al. [85] demonstrated that the agreement between standard dose and low dose (75% reduction) CAC score was significantly better for PCDs versus EIDs. Moreover, in the low-dose protocol, EIDs had the tendency to underestimate CAC scores, not present in PCDs.

It is well known that most acute myocardial infarctions are caused by occlusions in vessels with minor plaques that erode or rupture, the so-called ‘high-risk’ or ‘vulnerable’ atherosclerotic plaques [88]. Therefore, early identification of high-risk plaques may be useful for preventing ischemic events. The risk of atherosclerotic plaque rupture is primarily related to the composition of the plaques: vulnerable plaques display a large lipid-rich core, a thin fibrous cap, and an inflammatory infiltration [89]. Conventional CT comes with limitations in terms of a correct identification of plaque components. Inflammation is a component of all forms of plaque, but macrophages play a key role in acute plaque destabilization and thrombus formation. It has been shown that macrophages in atherosclerotic plaques of rabbits could be detected with conventional CT scanners after the intravenous injection of a contrast agent formed of iodinated nanoparticles dispersed with surfactant [90]. However, calcifications in the lumen wall could not be detected. In conventional CT imaging calcifications can interfere with macrophage burden quantification and a pre-injection CT scan would not be satisfactory because of strong artifacts surrounding the calcifications that could mask small focal enhancement. Several studies investigated the potential of PCCT K-edge imaging combined with gold nanoparticles for the contemporaneous assessment of different atherosclerosis aspects: measurement of lumen stenosis and characterization of plaque in terms of composition and vulnerability. Cormode et al. demonstrated in phantoms and apo E–KO mouse models of atherosclerosis that PCCT was able to accurately differentiate gold-based contrast agent, iodinated contrast agent, tissue, and calcium-rich matter, confirming the ability of the above-mentioned approach to detect macrophages in atherosclerosis while imaging the vasculature and calcified tissue [63]. Si-Mohamed et al. imaged atherosclerotic and control New Zealand white rabbits before and at 2 days after injection of gold nanoparticles and showed that the correlation between gold concentration and macrophage area was better using PCCT than conventional CT (0.82 vs. 0.41) [22]. Moreover, only gold K-edge PCCT allowed for the discrimination between enhancement of the lumen with one iodinated contrast material and enhancement of the vessel wall with K-edge gold nanoparticles. Importantly, the gold K-edge imaging findings were confirmed by histology (transmission electron microscopy and inductively coupled plasma optical emission spectrometry). It may be expected that, in virtue of the improved spatial resolution and the possibility to perform simultaneous material decomposition of multiple contrast agents, the PCCT would allow us to improve the evaluation of high-risk plaque features, adding value to the risk stratification of patients with CAD [91].

Coronary artery stenting is the most important non-surgical treatment for symptomatic coronary artery disease. Despite continuous improvements in stent design and medical treatment, in-stent restenosis (ISR) still remains a relevant issue after coronary stenting [92]. Therefore, the accurate reassessment of the vessel lumen after stent placement continues to be of paramount importance. The evaluation of stents and specifically of ISR with state-of-the-art EID technology is possible with good quality but continues to be partly impeded by metal artifacts, blooming, photon starvation, beam hardening, and partial volume effects [93]. Several studies investigated whether and how theoretical advantages of PCD yield better coronary stent imaging. In the in vitro study conducted by Mannil et al., 18 different coronary stents with different material composition were expanded in a plastic tube simulating the coronary artery and imaged with both PC and conventional CT systems, with identical settings for tube voltage and current, slice thickness, matrix size, and reconstruction kernel [94]. PCCT was associated with better delineation of lumen, lower image noise, reduction of blooming effect, and improved overall image quality. Other groups expanded on this study by implementing optimized convolution kernels which improved the spatial resolution of PCCT and better exploited the potential of PCDs. In the study by Symons et al., high-resolution (0.25 mm) PCCT resulted in superior coronary stent lumen visibility compared with standard resolution (0.5 mm) PCCT and conventional dual energy CT (EID technology) [95]. Furthermore, high-resolution PC acquisitions reconstructed at standard voxel size (0.5 mm isotropic) had lower image noise (25%) than standard-resolution PC acquisitions. In line with these findings, von Spiczak et al. found that the application of a sharp convolution kernel adapted to the intrinsic higher spatial resolution of the PCDs improved the coronary in-stent lumen visualization [96]. The resulting increase in image noise could be considerably reduced by using iterative image reconstruction techniques. Finally, Rajagopal et al. confirmed in their in vitro study that, compared to EID systems, HR-PCCT reconstructions enabled a more accurate evaluation of coronary lumen diameter [97]. They attributed this result to the fact that, besides the higher noise, high-resolution images had a more precise rendition of the high-contrast shape boundaries and lengths and were less affected by metal blooming artifacts. Additionally, all the three experienced radiologists who analyzed the images attributed to HR-PCCT images the higher plaque conspicuity and quality.

Feuerlein et al. created in a phantom a really challenging condition for conventional CCTA: a low-density calcified plaque located in a coronary metal stent with an attenuation level similar to that of the gadolinium-filled vascular lumen. The phantom was scanned with a preclinical PCCT using six energy thresholds. The performed gadolinium K-edge imaging resulted in a clear separation between calcified plaque and intravascular gadolinium and ineffective suppression of beam-hardening artifacts, allowing for an accurate characterization of the perfused vessel lumen [98].

As regards the noninvasive detection of ISR, the study by Bratke at al. was the first to demonstrate in vitro the clinically relevant role of PCCT [99]. Soft plaque-like restenosis were inserted into 10 different coronary stents, placed in the middle of plastic tubes (used as a coronary artery phantoms) and filled with a contrast agent. The image quality in terms of the visibility of the stenosis and the remaining lumen was judged significantly superior for PCCT compared to conventional CT. Stenosis was clearly detected in 9 and suspected in 10 of the 10 stents with both systems but the clear delineation of the residual lumen next to the stenosis was possible in 7 stents with PCCT and never possible using conventional CT.

### 5.2. Neuroimaging

Carotid artery atherosclerosis is considered a major risk factor for ischemic stroke and transient ischemic attacks (TIA) [100]. The accurate characterization of lumen narrowing together with plaque morphology are the key for an improved risk stratification of patients and the design of tailored medical therapy and surgical intervention [101]. CTA of the neck and brain vessels is routinely done in patients with suspected cerebrovascular stroke, but the calcified plaques leading to blooming and beam-hardening artifacts negatively impact the image quality [102]. In this context, all the added benefits of PCCT translate into improved imaging (Figure 3, Figure 4 and Figure 5).

The capability of PCCT technology to improve carotid and intracranial angiography has been demonstrated in vivo in a pilot study involving 16 asymptomatic subjects [49]. Compared to conventional CT, PCCT offered less image noise (9%) and beam-hardening artifacts in internal carotids close to the surrounding bone. Accordingly, the two radiologists blinded to the detector subsystem attributed to the images obtained with PCCT significantly higher quality scores for all vascular segments. Finally, the above-mentioned study showed the feasibility of spectral material decomposition of PCCT in the neck and brain for vascular imaging, although a direct comparison of the spectral performance between PCD and comparable dual energy EID CT scanners was not performed.

Hetterich et al. demonstrated in ex vivo (seven postmortem human carotid artery specimens) the high potential of PCCT for the quantitative evaluation of atherosclerotic carotid artery plaque [103]. When compared to histopathology, used as a standard reference, PCCT showed a good sensitivity and excellent specificity and accuracy for the detection of the necrotic core, fibrous cap, intraplaque hemorrhage, and calcifications. In addition, the correlation between PCCT and histopathology in terms of quantitative measurements of plaque components was excellent.

Sartoretti et al. investigated ex vivo (carotid artery specimen of deceased male donor) a preclinical PCCT scanner with an experimental tungsten-based contrast medium, characterized by higher atomic number and k-edge energies than iodine [104]. They demonstrated that the multi-energy bin option of PCCT combined with a spectrally optimized contrast medium was effectively advantageous in terms of calcium subtraction, allowing for a better visualization of the vessel lumen and atherosclerotic plaque compared with the standard iodine.

### 5.3. Abdominal Imaging

Figure 6 and Figure 7 show an abdominal CTA obtained with PCDs.

In the treatment of patients with abdominal aortic aneurysm (AAA), minimal invasive implantation of a covered, self-expandable stent graft—the endovascular aortic repair (EVAR)—represents a valid alternative to the more invasive conventional surgical repair [105,106]. The success of EVAR mainly depends on medical imaging and multislice CT is the preferred modality in both pre-operative planning and lifelong post-operative follow-up (1, 6, and 12 months after the intervention and annually thereafter) [107]. The endoleak is the most frequent complication of EVAR, typically requiring urgent interventions [108].

Due to the difficulty to distinguish between intra-aneurysmatic calcifications and leaking contrast media and the presence of different endoleaks flow rates, a triphasic CTA study is typically performed before and after the intravenous injection of iodinated contrast medium, with a significant radiation burden for the patient [109]. A single-phase, dual-energy CT with a split-bolus technique and the reconstruction of virtual non-enhanced images was demonstrated able to significantly reduce the radiation dose while maintaining a comparable endoleak detection rate [110]. However, with this approach, the type of endoleak (either low or high flow) cannot be precisely determined. Dangelmaier et al. showed in a phantom model that PCCT with the intravenous administration of two contrast agents (e.g., gadolinium and iodine) was able to capture endoleak dynamics and discriminate endoleaks from intra-aneurysmatic calcifications in a single scan, thereby allowing for a significant reduction of radiation exposure [111].

Sigovan et al. demonstrated the increased capacity of PCCT to detect correct stent deployment [112]. PC and conventional CT systems were used to image stents of different metal composition, deployed inside plastic tubes containing hydroxyapatite spheres to simulate vascular calcifications and in the abdominal aorta of one New Zealand White rabbit. In comparison with conventional CT, the increased spatial resolution of the PCCT enabled a better visualization of the intra-stent lumen, thanks to the significant reduction of stent-related blooming artifacts, as well as the morphological assessment of stent’s metallic mesh, even in the presence of calcification. Moreover, with PCCT the platinum-specific K-edge imaging enabled the exclusive visualization of the stent containing platinum only and the removal of other sources of attenuation.

**Table 1 diagnostics-13-00645-t001:** Studies on PCCT for cardio-vascular applications.

Reference	Type of Study	Main Finding
** *Coronary imaging* **		
Soschynski et al. 2022 [72]	Clinical (92 patients with chronic coronary syndrome).	Excellent imaging quality, very high CNR, and good assessability of coronary segments and vessels, even in cases of calcified plaques and stents, provided by PCCT.
Li et al. 2020 [77]	- Computer simulation. - In vitro-phantom (four phantoms with different stenosis severity, vessel diameters, and calcification densities).	Accuracy and precision of stenosis severity measurements higher in four-threshold PCCT images than DECT and two-threshold PCCT images.
Boussel et al. 2014 [80]	Ex vivo (10 calcified and 13 lipid-rich non-calcified plaques from post-mortem human coronary arteries).	Capability of PCCT to discriminate between calcifications and iodine-infused regions of human coronary artery atherosclerotic plaque samples, by detecting differences in spectral attenuation and iodine-based contrast agent concentration.
Si-Mohamed et al. 2022 [81]	- In vitro phantom (commercial phantom). - Clinical (14 patients with suspected or known CAD).	- Overall image quality, diagnostic quality, and diagnostic confidence in both calcified and non-calcified plaques significantly improved with PC compared to conventional dual-layer CT systems. - In comparison to EID-based CT images, 2.3- and 2.9-fold increased detectability index for coronary lumen and non-calcified plaque, respectively, achievable with PCCT.
Mergen et al. 2022 [82]	Clinical (20 patients with atherosclerotic plaques in proximal coronary arteries).	Reduced blooming artifacts with consequent improved visualization of fibrotic and lipid-rich plaque components obtained with the ultra-high-resolution mode of PCCT (slice thickness of 0.6 mm used as reference standard for comparison).
Sandstedt et al. 2021 [86]	Ex vivo (excised coronary specimens).	More accurate quantification of coronary calcifications and lower image noise achievable with high-resolution PCD-CT compared to conventional EID-CT.
van der Werf et al. 2022 [84]	In vitro phantom (anthropomorphic thorax phantom with inside CAC containing cylindrical inserts).	Improved CAC detection, even at 50% radiation dose reduction, and more accurate physical volume estimation, especially at reduced slice thickness and for high-density CAC, with PCCT compared to conventional CT.
van der Werf et al. 2022 [87]	In vitro phantom (anthropomorphic thorax phantom with artificial CAC with three densities).	Potential dose reduction of 50% for CAC scoring in medium- and high-density calcifications allowed by PCCT using low mono-energetic reconstructions.
Symons et al. 2019 [85]	- In vitro phantom (commercially available cardiac CT phantom). - Ex vivo (10 human hearts). - Clinical (10 asymptomatic volunteers).	Significant improvement in CAC score image quality or reduction of CAC score radiation, without a negative impact on diagnostic image quality, achievable with PCD compared to conventional EID CT.
Cormode et al. 2010 [63]	- In vitro phantom (phantoms containing gold high-density lipoprotein nanoparticle contrast agent, iodinated contrast agents, and calcium phosphate to simulate calcified tissue). - In vivo animal (mouse model of atherosclerosis).	Capability of PCCT to accurately differentiate gold-based contrast agent, iodinated contrast agent, tissue, and calcium-rich matter, which may allow for the simultaneous detection of macrophages in atherosclerosis and the imaging the vasculature and calcified tissue.
Si-Mohamed et al. 2021 [22]	In vivo animal (7 atherosclerotic and 4 control New Zealand white rabbits imaged before and after injection of gold nanoparticles).	- Better correlation between gold concentration and macrophage area using PCCT than conventional CT (0.82 vs. 0.41). - Simultaneous anatomic and molecular imaging of atherosclerosis allowed by PCCT k-edge imaging combined with gold nanoparticles.
Mannil et al. 2018 [94]	In vitro phantom (18 different coronary artery stents with different material composition, expanded in a plastic tube simulating the coronary artery).	Improved delineation of lumen, lower image noise, reduced blooming effect, and improved overall image quality with PCCT compared to conventional CT, despite the matched CT scan protocol settings and the identical image reconstruction parameters.
Symons et al. 2018 #110 [95]	In vitro phantom (18 coronary stents with different diameters implanted into a coronary artery phantom consisting of plastic tubes filled with contrast material).	- Superior coronary stent lumen visibility in high-resolution (0.25 mm) PCCT compared to standard resolution (0.5 mm) PCCT and conventional DECT. - Image noise decreased by 25% in high-resolution versus standard-resolution PCCT acquisitions.
von Spiczak et al. 2018 [96]	In vitro phantom (18 different coronary stents expanded in plastic tubes of 3 mm diameter, filled with diluted contrast agent, sealed, and immersed in oil).	Improved coronary in-stent lumen visualization with PCCT obtained thanks to the application of a sharp convolution kernel adapted to the intrinsic higher spatial resolution of the PCDs.
Rajagopal et al. 2021 [97]	In vitro (coronary artery phantom containing cylindrical probes simulating plaques with different composition and stenosis, imaged with and without coronary stents).	Improved visualization with less blooming artifacts and more accurate quantitative assessment of coronary plaques and stents with HR-PCCT compared to either photon-counting or energy-integrating CT.
Feuerlein et al. 2008. [98]	In vitro phantom (polymethylmethacrylate phantom with simulated low-density calcified plaque in a coronary metal stent).	Capability of gadolinium k-edge imaging performed with a multiple threshold–level PCCT to clearly separate the calcified plaque and the intra-vascular gadolinium and to effectively suppress the beam-hardening artifacts, for an accurate characterization of the perfused vessel lumen.
Bratke et al. 2020 [99]	In vitro phantom (10 different coronary stents placed in the middle of plastic tubes, used as a coronary artery phantoms and filled with a contrast agent).	- Superiority of PCCT compared to conventional CT for the noninvasive evaluation of ISR. - Clear delineation of the residual lumen next to the stenosis possible in 7/10 stents with PCCT and never possible using conventional CT.
** *Head and neck imaging* **		
Symons et al. 2018 [49]	Clinical (16 asymptomatic subjects).	- Better image quality, lower image noise, and less beam-hardening artifacts in internal carotids with PCCT compared to conventional CT. - Feasibility of spectral material decomposition of PCCT in the neck and brain for vascular imaging.
Hetterich et al. 2014 [103]	Ex vivo (7 postmortem human carotid artery specimens).	- Good sensitivity and excellent specificity and accuracy for the detection of the necrotic core, fibrous cap, intraplaque hemorrhage, and calcifications of PCCT versus histopathology, used as a standard reference. - Excellent correlation between PCCT and histopathology in terms of quantitative measurements of plaque components.
Sartoretti et al. 2020 [104]	Ex vivo (carotid artery specimen of deceased male donor).	Improved lumen and plaque visualization and image noise with PCCT employing the multi-energy bin option in combination with tungsten as contrast media compared with the standard iodine.
** *Abdominal imaging* **		
Dangelmaier et al. 2018 [111]	In vitro phantom (abdominal aortic aneurysm phantom filled with iodine, gadolinium, or calcium).	Ability of PCCT in combination with a dual contrast agent injection to capture endoleak dynamics and effectively distinct leaking contrast media from intra-aneurysmatic calcifications, thereby allowing for a significant reduction of radiation exposure.
Sigovan et al. 2019 [112]	- In vitro (stents of different metal composition deployed inside plastic tubes containing hydroxyapatite spheres to simulate vascular calcifications). - Animal (stents of different metal composition deployed in the abdominal aorta of one New Zealand White rabbit).	- Better visualization of the intra-stent lumen and morphological assessment of stent’s metallic mesh, even in the presence of calcification, with PCCT compared with conventional CT. - Exclusive visualization of the stent containing platinum achievable with PCCT with platinum-specific K-edge imaging.

## 6. Conclusions

PCCT offers wide ranging benefits over conventional CT, such as improved spatial and contrast resolution, significant noise reduction, dose efficiency and multi-energy capability. These key features have opened the door for a strongly improved performance of CT angiographic examinations while increasing patient safety. Certainly, larger patient cohort studies are still needed to identify and definitively prove the clinical impact of this new technology.

## Figures and Tables

**Figure 1 diagnostics-13-00645-f001:**
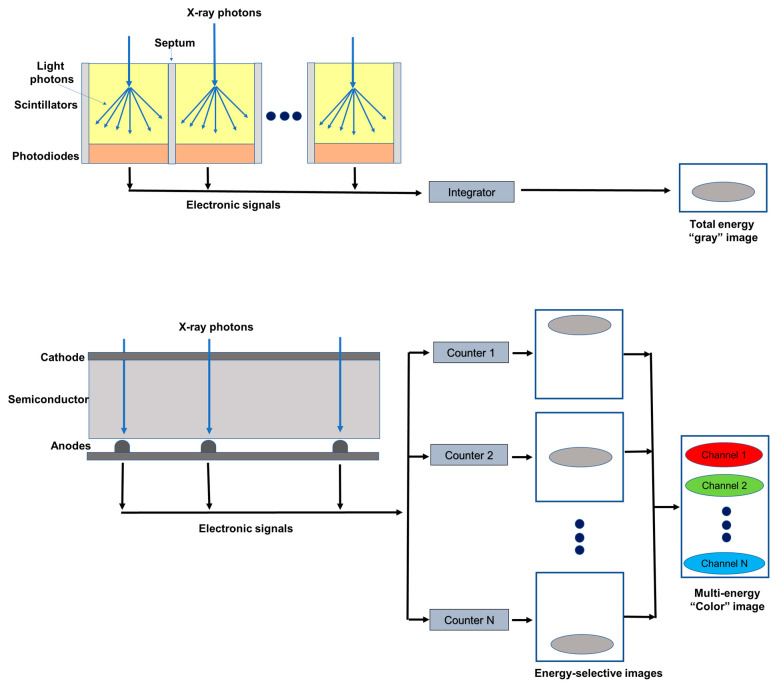
Schematic representation of an energy integrating detector (top) and of a photon-counting detector directly converting X-rays into an electrical signal (bottom). The photon-counting design allows the generation of energy-selective images, from which a set of material concentration maps can be obtained. Material concentration maps can then be combined in different ways to obtain monochromatic images, virtual non-contrast images, or material-specific color-overlay images.

**Figure 2 diagnostics-13-00645-f002:**
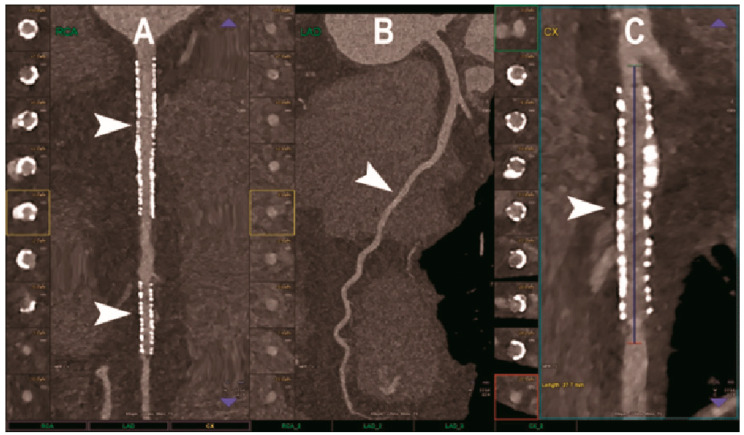
Cardiac CT using photon-counting computed tomography. The figure shows advanced multiplanar reconstructions of a coronary tree derived from a Photon-Counting CT (Scanner: NAEOTOM Alpha, Siemens) acquisition (**A**–**C**). In (**A**) we can see the right coronary artery with 2 stents (arrowheads), one proximal and one distal (much smaller), without any issue in the intrastent visualization of the arterial lumen. In (**B**), the left anterior descending coronary artery is depicted along its entire course (down to and beyond the left ventricular apex) with great detail and an evident deep intramyocardial course in the middle segment of the vessel (arrowhead). In (**C**), the left circumflex coronary artery also shows a stent with perfect intrastent visualization and patency (arrowhead).

**Figure 3 diagnostics-13-00645-f003:**
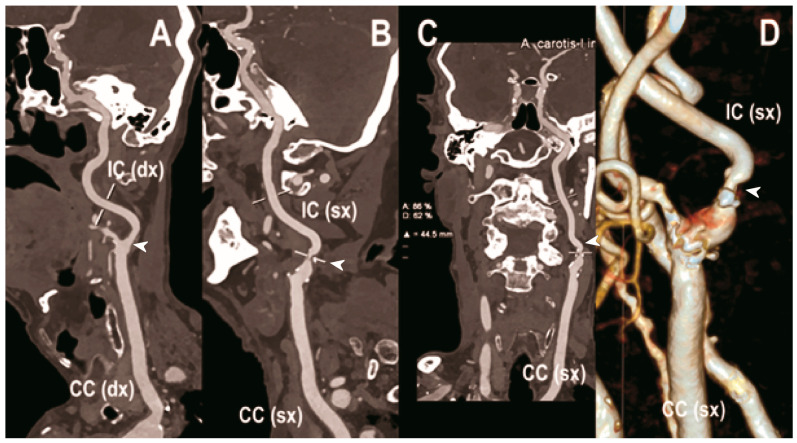
Carotid CT angiography using photon-counting computed tomography. The figure shows advanced reconstructions of a carotid artery tree derived from a photon-counting CT (Scanner: NAEOTOM Alpha, Siemens) acquisition (**A**–**D**). In (**A**), a longitudinal multiplanar reconstruction shows the right common and internal carotid artery with no significant luminal stenosis and a predominantly calcified atherosclerotic plaque at the carotid bifurcation (arrowhead). In (**B**), a longitudinal multiplanar reconstruction shows the left common and internal carotid artery with a significant luminal stenosis in the post-bulbar region of the internal carotid artery (arrowhead). In (**C**) the quantitative assessment of the stenosis (arrowhead) and in (**D**) the 3-dimensional volume rendering of the lesion (arrowhead).

**Figure 4 diagnostics-13-00645-f004:**
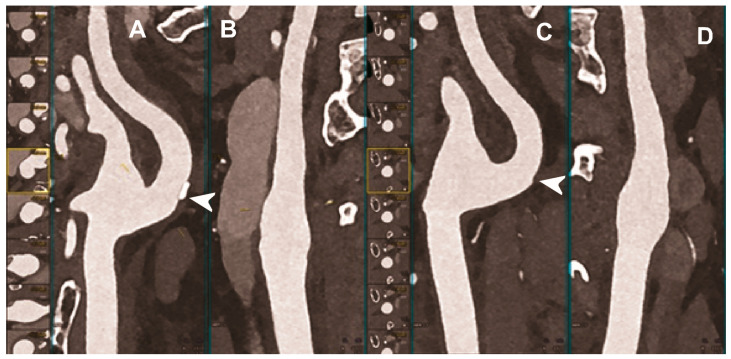
Carotid CT angiography using photon-counting computed tomography. The figure shows advanced reconstructions of a carotid artery tree derived from a photon-counting CT (Scanner: NAEOTOM Alpha, Siemens) acquisition (**A**–**D**). In (**A**,**B**), a longitudinal multiplanar reconstruction shows the right common and internal carotid artery with no significant luminal stenosis and a mild predominantly calcified atherosclerotic plaque at the internal carotid origin (arrowhead). In (**C**,**D**), a longitudinal multiplanar reconstruction shows the left common and internal carotid artery with a minimal arterial wall irregularity at the internal carotid origin (arrowhead). What is a bit unusual is to be able to see the thickness of the arterial wall at this level in a case with very mild atherosclerotic disease.

**Figure 5 diagnostics-13-00645-f005:**
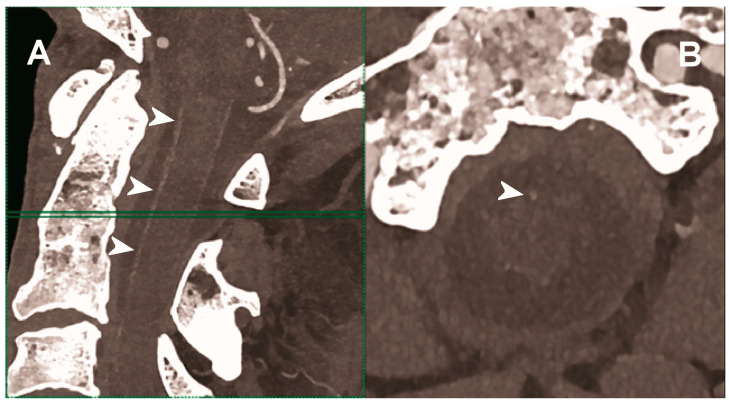
Carotid CT angiography using photon-counting computed tomography. The figure shows advanced MIP reconstructions of a intracervical artery tree derived from a photo-counting CT (Scanner: NAEOTOM Alpha, Siemens) acquisition. In (**A**) a sagittal median view of the cervical region showing the course of the anterior spinal artery (normally not visible) in the ventral portion of the rachidial channel (arrowheads). In (**B**) the axial image at the level of the green plane showed on the left panel with the axial view of the anterior spinal artery (arrowhead).

**Figure 6 diagnostics-13-00645-f006:**
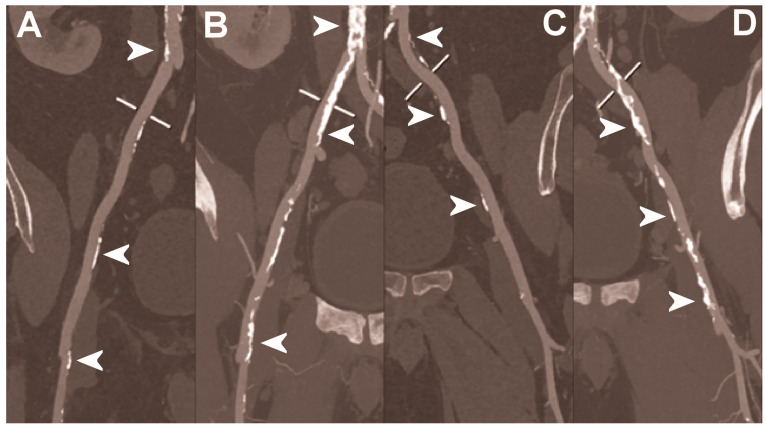
Abdominal CT angiography using photon-counting computed tomography. The figure shows advanced multiplanar reconstructions without and with MIP algorithm of a distal abdominal aorta and ilio-femoral arterial axes derived from a photon-counting CT (Scanner: NAEOTOM Alpha, Siemens) acquisition (**A**,**B** right; **C**,**D** left). The projection start in the abdominal aorta carrefour and end in the right/left common femoral artery. There are massive calcifications along the common iliac arteries; however, both MPRs (**A**,**C**) and MIPs (**B**,**D**) are so sharply defining the edges of the structures that lumen assessment is not compromised (arrowheads).

**Figure 7 diagnostics-13-00645-f007:**
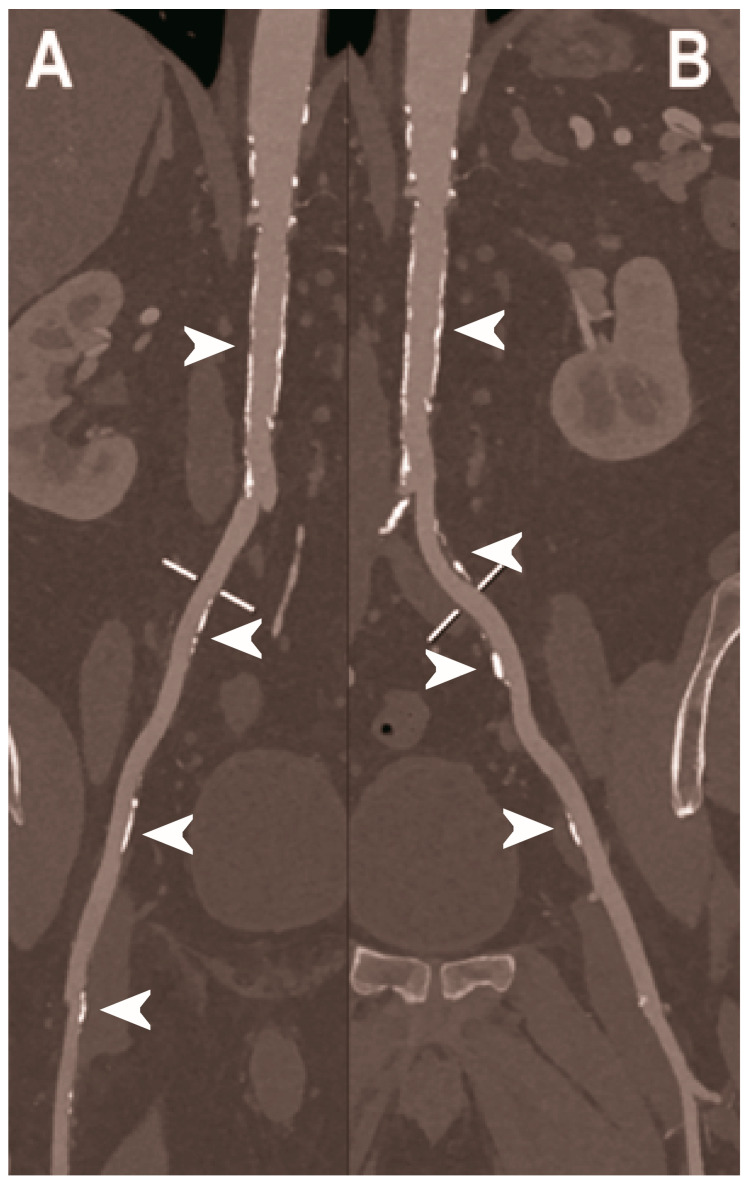
Abdominal CT angiography using photon-counting computed tomography. The figure shows advanced multiplanar reconstructions of an abdominal aorta and ilio-femoral arterial axes derived from a photon-counting CT (Scanner: NAEOTOM Alpha, Siemens) acquisition (**A**,**B**). In (**A**) the projection starts in the abdominal aorta at the level of thoraco-abdominal junction and ends in the right common femoral artery while in (**B**) it ends in the left common femoral artery. While there are significant calcifications along the vessels (arrowheads in **A** and **B**), the vessel wall is so sharp that the fact that there is no lumen reduction appears to be quite natural as compared to the common blooming effect seen with conventional energy-integrating detectors.

## Data Availability

Not applicable.

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
