# Peer review of "Photon-Counting Computed Tomography (PCCT): Technical Background and Cardio-Vascular Applications"

_diagnostics, 2023, doi:10.3390/diagnostics13040645_

Round 1

Reviewer 1 Report

Well written, timely review on photon counting CT in cardiovascular medicine. Carefully explained technical introduction and excellent clinical examples with well described applications.

Author Response

Well written, timely review on photon counting CT in cardiovascular medicine. Carefully explained technical introduction and excellent clinical examples with well described applications.

A: We would like to thank the Reviewer for the positive and really encouraging feedback.

Reviewer 2 Report

This article is well written. It reviews existing literature and summarizes the PCD/PCCT technology and its clinical application in cardiovascular imaging. However, I find it’s a little plain to read through, especially the figures are not very enlightening. I would encourage the authors to improve its presentation, expand the comparison, and then resubmit the manuscript.

Major comments:

* Please describe the literature selection process and how you identifies which articles to be included.

* Page 3, Figure 1: This figure needs further improvement.

- The text in the figure is very blurry and difficult to read.

- Please add the placement of “septa” in the figure-1 top row.

- Please explain in the main text or in the figure legend what the “color image” means.

- Use “multi-color” rather than 3 colors (and counters) as the energy level/threshold is not limited to 3.

* There are a lot of discussion and comparisons in the text for PCCT vs. DECT or EID-based CT. Please provide clinical examples in Figure 2, 3, 4, or phantom scans, for such comparisons and to show off the advantage of PCCT.

* Please compile a table for each of the subsections (CV, Neuro, Abdo) in section 4 to list the findings and comparisons of each reference as mentioned.

* Please add other relevant articles for cardiovascular imaging in the discussion. For example,

- Soschynski et al. DOI: 10.3390/jcm11206003

- Sandfort et al. doi:10.1016/j.jcct.2020.12.005

Also, please discuss Ultra-HR CT vs. PCCT for cardiovascular imaging

- Schuijf et al. doi:10.1016/j.jcct.2022.02.003

* Subsections 4.2 and 4.3 need more contents and better examples to show clinical examples and comparison.

* Please add a list of abbreviations.

Minor comments:

* Page 1, Line 30: Typo, please change “coronary heart disease (CAD)” to “coronary artery disease (CAD).

* Page 6 , Line 220: Please spell out DECT the first time it is used.

* Page 6 , Line 225: Two typos, reference # “56k”, and “PCTT”.

Author Response

This article is well written. It reviews existing literature and summarizes the PCD/PCCT technology and its clinical application in cardiovascular imaging. However, I find it’s a little plain to read through, especially the figures are not very enlightening. I would encourage the authors to improve its presentation, expand the comparison, and then resubmit the manuscript.

A: We would like to thank the Reviewer for the encouraging feedback and constructive critique and for the effort regarding this manuscript. We attempted to address each of the raised concerns, which  have substantially improved the manuscript.

Major comments:

* Please describe the literature selection process and how you identifies which articles to be included.

A: As now clarified in the text, this is not a systematic review and a PRISMA flow chart is not available. This is a narrative review, aimed to provide a critical overview of previously published research on PCCT for vascular applications.

A new section, entitled “Search strategy” has now been added to the main text as follows. “To prepare the narrative review we followed the indications present in [ref]. The article search was performed on PubMed, Scopus, and Google Scholar electronic databases between August and October 2022. We used the keywords “photon-counting computed tomography”, “PCCT”, “photon counting detector”, “photon counting X-ray detectors”, “photon counting CT”, “spectral CT”. Only articles written in English were included. Additional records identified through the list of references or other sources were also included. Two reviewers (AM and FF) analyzed the scientific papers to extract the relevant data for the purpose of this work.

* Page 3, Figure 1: This figure needs further improvement.

- The text in the figure is very blurry and difficult to read.

- Please add the placement of “septa” in the figure-1 top row.

- Please explain in the main text or in the figure legend what the “color image” means.

- Use “multi-color” rather than 3 colors (and counters) as the energy level/threshold is not limited to 3.

A: We have done in the Figure all the required modifications.

* There are a lot of discussion and comparisons in the text for PCCT vs. DECT or EID-based CT. Please provide clinical examples in Figure 2, 3, 4, or phantom scans, for such comparisons and to show off the advantage of PCCT.

A: We agree about the fact that there is a lot of comparison between DECT and PCCT, due to the fact that the majority of the published studies are comparative studies. Unfortunately, we don’t have the possibility of scanning the same patient with both CT scanners.

* Please compile a table for each of the subsections (CV, Neuro, Abdo) in section 4 to list the findings and comparisons of each reference as mentioned.

A: We have now added a Table that summarizes all studies.

* Please add other relevant articles for cardiovascular imaging in the discussion. For example,

- Soschynski et al. DOI: 10.3390/jcm11206003

- Sandfort et al. doi:10.1016/j.jcct.2020.12.005

We have now discussed the article of Soschynski et al. as follows. “The largest published clinical study of PCCT for CAD, involving 92 patients, demonstrated an excellent imaging quality, a very high CNR, and a good assessability of coronary segments and vessels, even in cases of calcified plaques and stents. Indeed, only 5% of the segments were rated non-diagnostic. The radiation dose was generally low and depended strongly on the scan mode. Nine patients underwent also invasive coronary angiography as the reference standard and the PCCT showed very high diagnostic performance for significant CAD on a per-segment level (sensitivity 92% and specificity 96%).”.

The review of Sandfort et al has now been cited.

Also, please discuss Ultra-HR CT vs. PCCT for cardiovascular imaging

- Schuijf et al. doi:10.1016/j.jcct.2022.02.003

A:  A brief Discussion on the UHR CT for cardiovascular imaging has now been added, as follows. “In this context, the high or ultra-high spatial resolution, achievable with both UHR EID-CT or PCCT scanners, could be particularly beneficial. Indeed, it conveys the potential for a more comprehensive evaluation of the coronary tree, a more precise grading of stenosis, and a better evaluation of segments with stents or extensive calcifications [ref].”

* Subsections 4.2 and 4.3 need more contents and better examples to show clinical examples and comparison.

A: As requested, we have now added more clinical examples.

* Please add a list of abbreviations.

 A list of abbreviations has now been added.

Minor comments:

* Page 1, Line 30: Typo, please change “coronary heart disease (CAD)” to “coronary artery disease (CAD).

A: We have done the required modification.

* Page 6 , Line 220: Please spell out DECT the first time it is used.

A: We have now spelled out DECT at the first mention

* Page 6 , Line 225: Two typos, reference # “56k”, and “PCTT”.

A: The typos have been corrected.

Reviewer 3 Report

In this article, the physical / technical background, and potential applications of PCCT are extensively described. The manuscript is interesting and relatively well written. Although studies on this issue are not present so far, it would be interesting to anticipate how the technical improvements provided by PCCT may improve the evaluation of high-risk plaque features, adding value to the risk stratification of patients with CAD and potentially rupture-prone plaques.

Otherwise, some minor comments can be found below:

Near universal availability may be a little bit exaggerated

The first paragraph of the introduction can be substantially shortened.

Author Response

We would like to thank the Reviewer for the encouraging feedback and constructive critique and for the effort regarding this manuscript. We have addressed each of the raised concerns, which have substantially improved the manuscript.

In this article, the physical / technical background, and potential applications of PCCT are extensively described. The manuscript is interesting and relatively well written. Although studies on this issue are not present so far, it would be interesting to anticipate how the technical improvements provided by PCCT may improve the evaluation of high-risk plaque features, adding value to the risk stratification of patients with CAD and potentially rupture-prone plaques.

A: We have now added in the text the following sentence: “It may be expected that, in virtue of the improved spatial resolution and the possibility to perform simultaneous material decomposition of multiple contrast agents, the PCCT would allow to improve the evaluation of high-risk plaque features, adding value to the risk stratification of patients with CAD.”.

Otherwise, some minor comments can be found below:

Near universal availability may be a little bit exaggerated

A: We have now used “large availability”.

The first paragraph of the introduction can be substantially shortened.

A: The first paragraph of the Introduction has been shortened.

Round 2

Reviewer 2 Report

Nice improvement of the revised manuscript.

Please fix picture formatting issues in Figure 1.

Consider adding arrowheads (as in Figure 2) to the Figure 3-7.

Author Response

Nice improvement of the revised manuscript.

A: We thank the Reviewer for the effort regarding this manuscript.

Please fix picture formatting issues in Figure 1.

A: Figure 1 has now been improved.

Consider adding arrowheads (as in Figure 2) to the Figure 3-7.

A: Arrowheads have now been added.

Reviewer 3 Report

The authors have been responsive to the comments of the reviewers. I therefore do not have any more queries with this paper.

Author Response

The authors have been responsive to the comments of the reviewers. I therefore do not have any more queries with this paper.

A: We thank the Reviewer for the effort regarding this manuscript.